# Learning with Missing Data: A Multimodal Hierarchical Variational Auto-Encoders for Medical Image Synthesis

**Reuben Dorent**[1] 🆔                                    REUBEN.DORENT@INRIA.FR
**Nazim Haouchine**[2] 🆔
**Alexandra Golby**[2] 🆔
**Sarah Frisken**[2] 🆔
**Tina Kapur**[2] 🆔
**Sandy Wells III**[2] 🆔

[1] *Sorbonne Université, CNRS, Inserm, AP-HP, Inria, Paris Brain Institute - ICM, Paris, France*
[2] *Harvard Medical School, Boston, USA*

## Abstract

Missing data is a widespread problem in multimodal medical imaging that presents significant practical and methodological challenges. While cross-modal synthesis has emerged as a promising strategy to estimate unavailable modalities from observed ones, most existing unified approaches assume complete multimodal datasets during training and rely on heuristic fusion strategies. In this short paper, we highlight the main ideas and findings of our recent work, where we introduced a Mixture of Multimodal Hierarchical Variational Auto-Encoders (MMHVAE) for unified cross-modal medical image synthesis from incomplete data. The model combines a hierarchical latent representation with a mixture of product-of-experts posterior to encode observed information, estimate missing information, and fuse arbitrary subsets of available modalities. The method was validated on the challenging problem of synthesis between multi-parametric MRI and intraoperative ultrasound in brain tumor patients, with additional evaluation on two downstream tasks: segmentation and registration. This work shows that principled probabilistic models can learn rich and informative multimodal representations from incomplete imaging datasets.

**Keywords:** Missing Data, Hierarchical Variational Autoencoders, Image Synthesis

## 1. Introduction

Different sensors collect patient data at different stages of the clinical workflow, including diagnosis, monitoring, surgery, and follow-up. However, multimodal data is expensive and sparse, resulting in incomplete image sets. For example, some MR sequences are typically acquired before brain surgery, while ultrasound data can only be acquired during surgery, when MR acquisition is impractical. Moreover, various clinical issues (e.g., technical failure, data loss, patient issues) can lead to a lack of data. There is thus a strong need to develop methods capable of handling incomplete multimodal data at both training and test time.

To tackle missing imaging data, deep-learning methods based on generative adversarial networks (Li et al., 2019), transformers (Dalmaz et al., 2022), and diffusion models (Meng et al., 2024) have been proposed to perform cross-modal image synthesis, where missing images are estimated from observed images in other modalities. Specifically, these unified approaches are trained using artificially zero- or noise-imputed data. However, they do not explicitly learn a shared multimodal representation and assume access to complete

data during training, limiting their ability to leverage incomplete observed datasets. In contrast, Multimodal Variational Auto-Encoders (VAEs) (Wu and Goodman, 2018; Shi et al., 2019; Sutter et al., 2021) provide a principled formulation for learning a common latent space across modalities. Yet, they rely on low-dimensional latent representations that are not well suited to high-resolution image synthesis, and also assume complete training sets. Hierarchical VAEs (Ranganath et al., 2016), on the other hand, have demonstrated improved generative performance by leveraging more expressive latent representations, but these methods are mono-modal and do not handle missing data.

In this paper, we summarize our recent work (Dorent et al., 2026), which addresses this critical problem by proposing a hierarchical multimodal VAE that learns rich and informative multimodal latent representations directly from incomplete data. The work combines complex probabilistic graphical models with advanced deep learning architectures, closing the gap between principled formulations and state-of-the-art synthesis performance.

## 2. Mixture of Multimodal Hierarchical Variational Auto-Encoders

The main contribution of our work (Dorent et al., 2026) is a principled framework to encode $M$ multimodal images $(x_i)_{i=1}^M$ into a shared hierarchical latent representation $z$, from which missing images can be synthesized from any subset of observed data. More specifically, the method addresses four key challenges: (i) creating a complex latent representation of multimodal data to generate high-resolution images; (ii) encouraging the variational distributions to estimate the missing information needed for cross-modal image synthesis; (iii) learning to fuse multimodal information in the context of missing data; and (iv) leveraging dataset-level information to handle incomplete datasets during training.

The model is built as a hierarchical multimodal VAE, where the latent variable $z$ is partitioned into multiple levels $(z_l)_{l=1}^L$. This hierarchical design allows to model image variability at different spatial scales, from global descriptors to local pixel-level features. In contrast to multimodal VAEs, which produce blurry synthetic images, the hierarchical representation enables much detailed synthesis.

A second contribution is to model the variational posterior as a mixture of product-of-experts. Each product-of-experts component has a factorization similar to the true posterior and can be computed from any subset of observed modalities using only unimodal encoders. This probabilistic formulation not only provides flexible fusion for incomplete observations, it also leads to an evidence lower bound that is maximal when the variational posterior distributions align with the complete true posterior, i.e. approximate posteriors should encode the observed information while estimating the missing one.

Finally, unlike existing unified synthesis frameworks, data missingness, modeled by the random variable $R$, is expected at training time. To regularize image distributions when data is partially observed, an adversarial strategy that leverages dataset-level information is used.

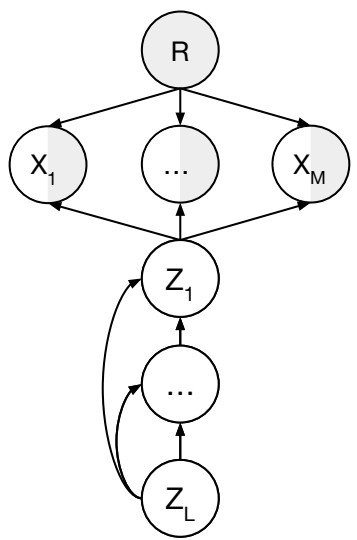

Figure 1: Graphical model of MMHVAE.

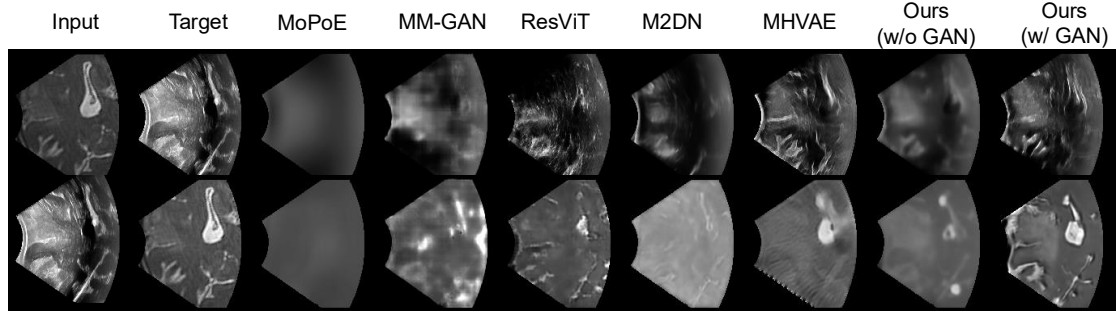

| Input | Target | MoPoE | MM-GAN | ResViT | M2DN | MHVAE | Ours (w/o GAN) | Ours (w/ GAN) |

Figure 2: Qualitative comparison of our method with all competing methods

## 3. Experimental validation

The method was validated on the challenging problem of cross-modal synthesis between multi-parametric MRI ($T_2$, $ceT_1$, and FLAIR) and intraoperative ultrasound (iUS) in brain tumor patients. The ReMIND (Juvekar et al., 2024) dataset ($N = 114$), which is highly incomplete (87% of cases missing at least one modality), was used as a realistic benchmark for learning with missing multimodal data. We compared our approach with state-of-the-art unified methods, including the GAN-based MM-GAN (Sharma and Hamarneh, 2020), the transformer-based ResViT, and the diffusion-based M2DN.

**Cross-modal image synthesis.** MMHVAE was first assessed on harmonized cross-modal synthesis from arbitrary subsets of inputs. Our method consistently outperforms competing approaches across all modalities and three synthesis performance measurements (PSNR, SSIM, LPIPS). Moreover, our approach is significantly more computationally efficient than the best-performing baseline, ResViT, with a much lower time complexity (13G vs. 274G MACs) and a smaller model size (14M vs. 293M parameters). Compared to M2DN, our method is both faster (55 ms vs. 290 ms) and more efficient (13G vs. 24G MACs).

**Ablation studies.** Ablation experiments confirmed the importance of our contributions. First, increasing the depth of the hierarchical latent representation substantially improved synthesis quality, highlighting the value of modeling image variability at multiple scales. As $L$ increases from 1 to 7 PSNR improves by 6.2,dB for $T_2$ and 7.6,dB for FLAIR. Second, the proposed probabilistic fusion consistently outperformed simpler strategies such as zero-imputation and feature averaging. Finally, GAN-based regularization improved perceptual realism, especially for ultrasound, reducing perceptual LPIPS from 20.6% to 14.6%.

**Downstream tasks.** Evaluation included two downstream tasks: brain tumor segmentation in iUS and MR–iUS registration. Segmentation models trained on our iUS synthetic data outperformed those trained with competing methods, reaching mean Dice scores of 73.6% on RESECT-SEG (Behboodi et al., 2022) and 77.6% on ReMIND, versus 67.3% and 74.7% for the strongest baseline. Synthetic MR images also improved registration over direct MR–iUS registration: in the 8–12 mm displacement range, target registration error fell to 3.2 mm ($ceT_1$) and 4.4 mm (FLAIR) compared with 14.6 mm and 9.4 mm using iUS.

These results demonstates that principled probabilistic models can learn from incomplete imaging datasets and perform competitive medical image synthesis in real clinical settings.

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

Table 1: **Comparison against the state-of-the-art unified models for image synthesis**, including MoPoE (Sutter et al., 2021), MHVAE (Dorent et al., 2023), MM-GAN (Sharma and Hamarneh, 2020), ResViT (Dalmaz et al., 2022) and M2DN (Meng et al., 2024). Mean and standard deviation values are presented. For a given input, values in bold indicate the methods that achieved the best performance and were found to be statistically significantly better than all others, based on a paired Wilcoxon signed-rank test with Bonferroni correction ($p < 0.01$). Arrows indicate the favorable direction of each metric.

| | Input | | | | iUS | | | T$_2$ | | | ceT$_1$ | | | FLAIR | | |
|---|---|---|---|---|---|---|---|---|---|---|---|---|---|---|---|---|
| | iUS | T$_2$ | T$_1$ | FL | PSNR(dB)↑ | SSIM(%)↑ | LPIPS(%)↓ | PSNR(dB)↑ | SSIM(%)↑ | LPIPS(%)↓ | PSNR(dB)↑ | SSIM(%)↑ | LPIPS(%)↓ | PSNR(dB)↑ | SSIM(%)↑ | LPIPS(%)↓ |
| MoPoE | ● | ○ | ○ | ○ | 22.4 (3.6) | 75.7 (11.7) | 31.6 (13.1) | **21.8 (5.0)** | **83.0 (8.6)** | 24.1 (10.5) | **23.6 (5.3)** | **84.9 (7.6)** | 20.2 (9.5) | 18.4 (8.2) | **79.3 (10.3)** | 22.9 (11.4) |
| MHVAE | ● | ○ | ○ | ○ | 31.6 (2.2) | 92.4 (3.3) | 6.8 (2.6) | 21.6 (4.0) | 80.7 (9.1) | 20.8 (8.8) | - | - | - | - | - | - |
| MM-GAN | ● | ○ | ○ | ○ | 25.3 (5.0) | 87.3 (7.3) | 10.8 (5.6) | 19.5 (4.6) | 76.7 (11.0) | 19.9 (8.2) | 21.3 (4.6) | 79.5 (9.4) | 17.0 (7.8) | 16.7 (7.7) | 74.3 (11.9) | 19.3 (9.7) |
| ResViT | ● | ○ | ○ | ○ | 32.5 (4.1) | 95.6 (2.6) | 3.2 (1.7) | 20.7 (4.2) | 79.1 (9.8) | **16.0 (6.7)** | 22.5 (5.0) | 81.7 (8.4) | **13.5 (6.6)** | 17.8 (7.5) | 76.5 (10.8) | **14.5 (9.3)** |
| M2DN | ● | ○ | ○ | ○ | 27.7 (5.5) | 88.0 (7.1) | 5.5 (2.8) | 19.4 (5.1) | 80.1 (9.5) | 20.1 (8.9) | 20.7 (5.7) | 83.0 (8.1) | 17.1 (8.3) | 18.3 (7.5) | 78.6 (10.2) | 16.9 (10.0) |
| Ours | ● | ○ | ○ | ○ | **46.3 (2.9)** | **99.6 (0.5)** | **0.2 (0.4)** | 20.8 (4.3) | 79.9 (9.6) | 16.3 (6.6) | 22.2 (5.4) | 82.0 (8.5) | **14.0 (6.8)** | 18.6 (8.0) | 79.0 (10.2) | 17.2 (9.5) |
| MoPoE | ○ | ● | ○ | ○ | **21.0 (4.4)** | **74.4 (12.4)** | 31.6 (13.7) | 21.8 (5.1) | 83.2 (8.6) | 24.0 (10.5) | 24.4 (4.5) | 85.9 (6.9) | 19.3 (9.2) | 20.3 (4.8) | 81.0 (9.2) | 21.4 (9.9) |
| MHVAE | ○ | ● | ○ | ○ | 20.8 (3.7) | 73.9 (12.0) | 20.5 (8.5) | 28.6 (3.1) | 91.0 (3.9) | 5.7 (2.1) | - | - | - | - | - | - |
| MM-GAN | ○ | ● | ○ | ○ | 20.5 (4.4) | 73.3 (12.5) | 25.4 (10.6) | 23.3 (4.4) | 93.0 (3.4) | 5.4 (2.1) | 24.4 (3.5) | 86.1 (6.4) | 10.6 (4.9) | 19.8 (3.8) | 79.3 (9.7) | 12.5 (5.4) |
| ResViT | ○ | ● | ○ | ○ | 20.0 (4.2) | 70.9 (13.3) | 19.3 (7.9) | 28.6 (4.6) | 94.6 (3.0) | 5.0 (2.7) | **24.9 (4.1)** | **86.9 (6.4)** | **8.8 (4.5)** | 22.4 (3.4) | 83.5 (7.7) | **8.5 (3.8)** |
| M2DN | ○ | ● | ○ | ○ | 20.6 (4.9) | **74.5 (12.4)** | 23.2 (9.3) | 24.9 (5.3) | 95.0 (3.1) | 4.6 (2.5) | 20.6 (4.8) | 84.9 (7.3) | 15.4 (7.5) | 21.3 (4.4) | 82.1 (8.6) | 12.9 (6.1) |
| Ours | ○ | ● | ○ | ○ | 20.6 (4.2) | 73.1 (12.4) | 18.9 (7.6) | **35.9 (4.1)** | **98.5 (1.2)** | **1.1 (0.7)** | 24.9 (4.3) | 86.7 (6.5) | 9.0 (4.4) | 23.5 (4.1) | 85.1 (7.3) | 9.0 (4.4) |
| MoPoE | ○ | ○ | ● | ○ | 20.9 (4.5) | 73.8 (12.4) | 32.3 (13.6) | 21.7 (5.0) | 82.8 (8.6) | 24.3 (10.5) | 23.6 (5.1) | 85.0 (7.5) | 20.4 (9.5) | 18.3 (8.2) | 79.2 (10.0) | 23.9 (11.0) |
| MM-GAN | ○ | ○ | ● | ○ | 20.0 (4.2) | 72.4 (12.6) | 26.3 (10.6) | 21.2 (4.5) | 82.1 (8.5) | 13.6 (5.5) | 28.1 (4.0) | 94.8 (3.0) | 4.4 (2.2) | 18.6 (6.7) | 78.1 (10.3) | 14.3 (7.6) |
| ResViT | ○ | ○ | ● | ○ | 19.8 (4.2) | 70.0 (13.5) | 19.8 (8.0) | 22.6 (4.4) | 82.6 (8.2) | 12.1 (5.2) | 29.6 (5.4) | 94.7 (3.1) | 4.3 (3.5) | 20.1 (7.9) | 81.6 (8.5) | **10.6 (8.5)** |
| M2DN | ○ | ○ | ● | ○ | 19.8 (5.0) | 73.1 (12.9) | 24.8 (9.7) | 19.3 (5.3) | 80.4 (9.4) | 20.3 (9.0) | 24.7 (7.5) | 93.5 (3.8) | 6.7 (3.9) | 18.5 (8.0) | 78.9 (10.0) | 17.4 (10.2) |
| Ours | ○ | ○ | ● | ○ | **21.1 (4.2)** | **74.0 (11.9)** | 20.9 (8.8) | **23.2 (4.4)** | **84.2 (7.9)** | **10.5 (4.6)** | **36.1 (6.2)** | **98.6 (2.0)** | **1.3 (3.0)** | 21.4 (7.8) | **83.6 (7.9)** | **10.5 (7.3)** |
| MoPoE | ○ | ○ | ○ | ● | **20.9 (4.5)** | **74.8 (12.6)** | 31.2 (13.8) | 22.0 (4.9) | **84.1 (8.0)** | 23.3 (10.1) | 22.5 (7.2) | 84.9 (7.8) | 20.8 (10.2) | 18.3 (7.8) | 79.5 (9.7) | 23.9 (11.5) |
| MM-GAN | ○ | ○ | ○ | ● | 20.5 (4.3) | 73.5 (12.8) | 25.4 (11.0) | 20.5 (4.3) | 80.3 (9.4) | 15.9 (6.9) | 22.4 (7.0) | 83.1 (8.4) | 13.4 (7.9) | 22.2 (7.2) | 86.9 (7.3) | 9.3 (6.9) |
| ResViT | ○ | ○ | ○ | ● | 20.0 (4.3) | 71.3 (13.5) | **19.2 (8.0)** | 22.6 (4.2) | 83.4 (8.0) | **11.5 (5.0)** | 22.7 (7.9) | **85.2 (7.2)** | **10.6 (6.8)** | 27.2 (8.6) | 95.0 (5.2) | 4.2 (8.3) |
| M2DN | ○ | ○ | ○ | ● | 20.6 (5.0) | **74.8 (12.7)** | 22.8 (9.3) | 18.6 (5.2) | 81.2 (9.0) | 18.4 (8.1) | 18.5 (8.2) | 83.3 (8.2) | 16.8 (8.4) | 22.0 (8.7) | 94.5 (4.7) | 4.5 (6.3) |
| Ours | ○ | ○ | ○ | ● | 20.3 (4.4) | 73.4 (12.9) | 20.7 (9.1) | 21.7 (4.4) | 83.6 (8.1) | 11.9 (4.8) | 22.4 (7.5) | 84.1 (7.9) | 10.9 (7.4) | **31.3 (8.6)** | **97.6 (5.1)** | **2.5 (7.6)** |
| MoPoE | ○ | ● | ● | ○ | **21.0 (4.5)** | **74.3 (12.4)** | 31.9 (13.6) | 21.8 (5.0) | 82.9 (8.6) | 24.3 (10.5) | 24.4 (4.4) | 85.9 (6.9) | 19.3 (9.2) | 20.2 (4.9) | 80.3 (9.3) | 22.1 (9.9) |
| MM-GAN | ○ | ● | ● | ○ | 20.7 (4.4) | 73.6 (12.4) | 24.9 (10.4) | 24.0 (4.4) | 93.3 (3.3) | 5.1 (2.1) | 28.2 (3.1) | 93.9 (2.7) | 5.0 (2.0) | 20.4 (3.8) | 80.6 (9.3) | 12.0 (5.3) |
| ResViT | ○ | ● | ● | ○ | 20.1 (4.3) | 71.1 (13.2) | 19.0 (7.8) | 28.2 (4.7) | 93.9 (3.3) | 5.2 (2.9) | 29.2 (3.9) | 93.6 (3.2) | 4.6 (2.6) | 22.8 (3.4) | 84.6 (7.2) | **7.6 (3.5)** |
| M2DN | ○ | ● | ● | ○ | 20.6 (4.8) | **74.4 (12.2)** | 23.0 (9.2) | 24.9 (5.4) | 95.0 (3.0) | 4.5 (2.3) | 24.9 (7.2) | 93.9 (3.7) | 5.4 (2.7) | 21.6 (4.3) | 82.7 (8.4) | 12.1 (5.7) |
| Ours | ○ | ● | ● | ○ | **20.9 (4.0)** | 73.6 (12.0) | 18.4 (7.4) | **35.6 (4.5)** | **98.4 (1.3)** | **1.1 (0.7)** | **36.1 (4.6)** | **98.7 (0.7)** | **1.1 (0.6)** | **23.9 (3.8)** | **85.8 (6.9)** | 8.6 (4.2) |
| MoPoE | ○ | ● | ○ | ● | **21.3 (4.6)** | 75.5 (12.5) | 30.9 (14.0) | 22.3 (4.9) | 84.3 (8.0) | 23.3 (10.1) | 24.2 (4.4) | 86.3 (6.6) | 18.5 (8.5) | 20.2 (4.7) | 80.9 (9.2) | 21.4 (9.8) |
| MM-GAN | ○ | ● | ○ | ● | 21.1 (4.5) | 74.9 (12.3) | 24.4 (10.9) | 24.1 (4.2) | 93.3 (3.3) | 4.7 (2.0) | 24.9 (3.5) | 87.1 (6.1) | 9.3 (4.3) | 25.7 (4.2) | 91.2 (4.6) | 5.5 (2.5) |
| ResViT | ○ | ● | ○ | ● | 20.3 (4.3) | 72.4 (13.2) | 18.5 (7.9) | 28.3 (4.5) | 94.0 (3.2) | 5.3 (3.0) | **25.1 (3.9)** | **88.2 (5.9)** | **7.8 (3.8)** | 28.3 (3.6) | 95.2 (2.4) | 3.2 (2.1) |
| M2DN | ○ | ● | ○ | ● | **21.2 (5.0)** | **75.8 (12.4)** | 21.7 (9.1) | 25.5 (5.1) | 95.5 (2.9) | 3.8 (2.0) | 19.8 (5.3) | 85.8 (7.0) | 13.6 (6.1) | 22.6 (6.9) | 95.2 (2.6) | 3.4 (1.9) |
| Ours | ○ | ● | ○ | ● | 20.4 (4.0) | 73.7 (12.5) | 18.1 (7.6) | **35.1 (4.2)** | **98.3 (1.5)** | **1.4 (0.9)** | 24.6 (3.9) | 87.0 (6.1) | 8.8 (4.0) | **33.3 (4.0)** | **98.2 (1.1)** | **1.3 (0.8)** |
| MoPoE | ○ | ○ | ● | ● | 20.9 (4.6) | **74.4 (12.6)** | 31.8 (13.9) | 22.0 (4.7) | 83.6 (8.1) | 24.1 (10.1) | 22.8 (7.0) | 85.2 (7.8) | 20.3 (8.8) | 18.2 (7.9) | 79.1 (9.7) | 24.3 (10.4) |
| MM-GAN | ○ | ○ | ● | ● | 20.5 (4.2) | 73.4 (12.6) | 25.3 (10.7) | 21.4 (4.3) | 82.2 (8.2) | 14.9 (6.3) | 26.9 (6.8) | 93.2 (5.8) | 5.3 (5.8) | 23.0 (7.5) | 89.0 (6.7) | 7.7 (6.9) |
| ResViT | ○ | ○ | ● | ● | 20.0 (4.3) | 71.2 (13.3) | **19.1 (7.9)** | 23.4 (4.4) | 84.7 (7.3) | 10.4 (4.7) | 27.2 (8.6) | 93.1 (4.5) | 5.6 (6.3) | 26.3 (9.0) | 94.2 (5.5) | 4.7 (8.8) |
| M2DN | ○ | ○ | ● | ● | 20.6 (4.9) | **74.4 (12.5)** | 22.9 (9.2) | 18.5 (5.2) | 81.7 (8.7) | 17.1 (7.4) | 22.7 (9.4) | 93.7 (4.8) | 5.7 (6.0) | 22.4 (9.0) | 94.6 (4.9) | 4.5 (6.8) |
| Ours | ○ | ○ | ● | ● | **20.9 (4.0)** | 74.0 (12.0) | 19.4 (8.3) | **23.9 (4.0)** | **85.6 (6.9)** | **9.5 (3.8)** | **33.5 (9.8)** | **97.9 (4.0)** | **2.1 (6.2)** | **31.1 (8.9)** | **97.6 (4.0)** | **2.4 (6.7)** |
| MoPoE | ○ | ● | ● | ● | **21.2 (4.6)** | **75.0 (12.4)** | 31.5 (13.9) | 22.1 (4.9) | 83.6 (8.1) | 24.0 (10.1) | 24.3 (4.3) | 86.4 (6.6) | 18.5 (8.4) | 20.2 (4.7) | 80.3 (9.2) | 22.0 (9.8) |
| MM-GAN | ○ | ● | ● | ● | 21.1 (4.4) | 74.6 (12.2) | 24.4 (10.7) | 25.3 (4.0) | 93.8 (3.0) | 4.5 (1.9) | 28.1 (3.5) | 93.6 (3.0) | 4.8 (2.0) | 26.2 (3.9) | 92.2 (4.1) | 5.2 (2.4) |
| ResViT | ○ | ● | ● | ● | 20.3 (4.3) | 72.1 (13.0) | 18.6 (7.8) | 27.9 (4.7) | 93.5 (3.4) | 5.6 (3.2) | 28.2 (3.9) | 93.0 (3.5) | 4.8 (2.4) | 27.8 (3.5) | 94.7 (2.7) | 3.5 (2.2) |
| M2DN | ○ | ● | ● | ● | 21.0 (4.9) | **75.1 (12.3)** | 22.0 (9.0) | 25.9 (5.0) | 95.5 (2.5) | 3.9 (1.9) | 23.6 (6.9) | 94.3 (3.5) | 4.7 (2.4) | 23.2 (6.7) | 95.4 (2.3) | 3.4 (1.8) |
| Ours | ○ | ● | ● | ● | 20.6 (3.9) | 73.9 (12.1) | 18.0 (7.4) | **35.1 (4.3)** | **98.2 (1.6)** | **1.3 (0.8)** | **34.2 (5.0)** | **98.4 (1.0)** | **1.2 (0.7)** | **32.8 (4.3)** | **98.1 (1.1)** | **1.5 (0.8)** |
| _Average_ MoPoE | | | | | 21.3 (4.3) | 74.6 (12.3) | 31.7 (13.5) | 21.8 (5.0) | 83.2 (8.5) | 24.1 (10.4) | 23.8 (5.2) | 85.4 (7.3) | 19.8 (9.3) | 19.2 (6.8) | 79.9 (9.6) | 22.8 (10.6) |
| MM-GAN | | | | | 21.6 (4.9) | 76.7 (13.0) | 21.9 (11.5) | 22.2 (4.8) | 86.5 (10.0) | 10.8 (7.9) | 25.4 (5.2) | 88.5 (8.8) | 9.3 (7.2) | 21.4 (6.8) | 83.7 (10.6) | 11.0 (7.9) |
| ResViT | | | | | 23.0 (6.8) | 76.9 (15.8) | 15.4 (9.8) | 25.1 (5.5) | 87.8 (9.2) | 9.3 (6.3) | 26.3 (5.9) | 89.1 (7.8) | 7.9 (6.1) | 24.0 (7.5) | 88.0 (10.0) | 7.2 (7.8) |
| M2DN | | | | | 22.2 (6.0) | 77.5 (12.9) | 19.1 (11.3) | 22.1 (6.0) | 87.8 (10.1) | 12.0 (10.0) | 22.4 (7.0) | 88.8 (7.9) | 11.1 (8.1) | 21.1 (7.5) | 87.7 (10.3) | 9.5 (9.0) |
| Ours | | | | | **27.0 (11.6)** | **79.9 (15.4)** | **14.8 (10.9)** | **28.9 (8.0)** | **90.5 (10.1)** | **7.0 (7.3)** | **29.5 (8.5)** | **91.4 (9.2)** | **6.4 (7.2)** | **26.9 (8.7)** | **90.5 (9.9)** | **6.8 (8.3)** |

# 4. Appendices

## 4.1. Implementation details

**Network Architecture:** Given that raw brain ultrasound images are typically 2D, we adopted a 2D U-Net-based architecture. The spatial resolution and feature dimension of the coarsest latent variable ($z_L$) were chosen as $1 \times 1$ and 256, respectively. The spatial and feature dimensions were successively doubled and halved after each level, resulting in a

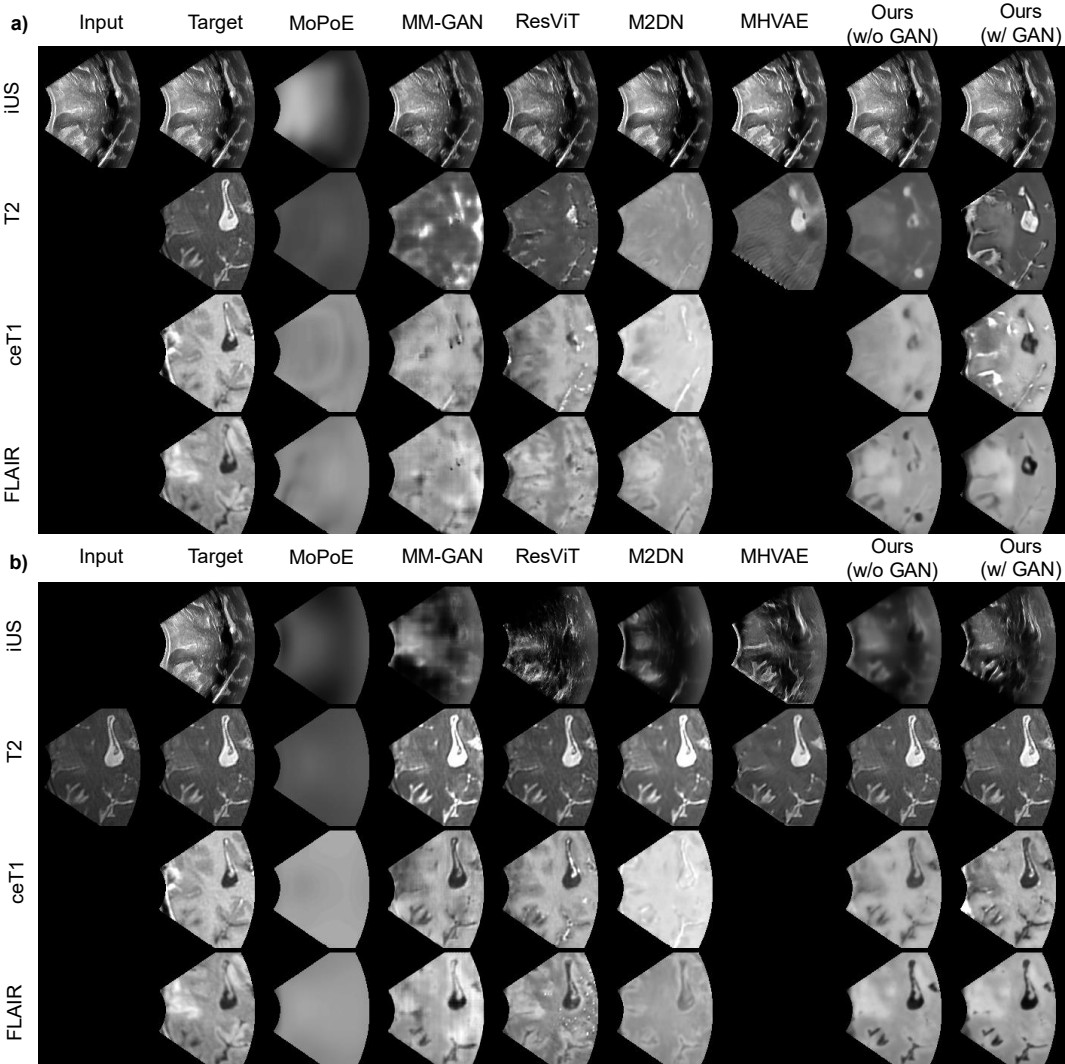

Figure 3: Qualitative comparison of our method with all competing methods for synthesizing all modalities (iUS, $T_2$, $ceT_1$, FLAIR) from (a) iUS; (b) $T_2$. Our approach generates sharper images with better contrast differentiation between tissues and modality-specific patterns (e.g. speckles for iUS).

feature representation of dimension 8 for each pixel at group 1, denoted as $z_1 \in \mathbb{R}^{192 \times 192 \times 8}$. This results in $L = 7$ latent variable levels. Following state-of-the-art architectures (Vahdat and Kautz, 2020), residual cells from MobileNetV2 (Sandler et al., 2018) are employed for the encoder and decoder, with Squeeze and Excitation (Hu et al., 2018) and Swish activation. The image decoders $(\mu_{\theta_j^x})_{j=1}^M$ correspond to 5 ResNet blocks. At inference, we lower the temperature of the parametric distributions to 0.5, as performed in other HVAEs(Vahdat and Kautz, 2020). The code is available at https://github.com/ReubenDo/MMHVAE.

Table 2: **Comparison against the state-of-the-art unified models on brain tumor segmentation downstream tasks in iUS.** Best performance in bold is based on a paired Wilcoxon signed-rank test ($p < 0.05$).

| | RESECT-SEG (Behboodi et al., 2022) | | ReMIND (Juvekar et al., 2024) | |
|---|---|---|---|---|
| | Dice Score (%)↑ | ASSD (mm)↓ | Dice Score (%)↑ | ASSD (mm)↓ |
| MM-GAN (Sharma and Hamarneh, 2020) | 53.3 [37.1 - 69.8] | 4.1 [3.4 - 5.3] | 57.9 [52.6 - 71.5] | 5.0 [3.3 - 6.0] |
| ResViT (Dalmaz et al., 2022) | **67.3 [48.3 - 80.5]** | **2.3 [1.7 - 3.7]** | **74.7 [69.0 - 79.6]** | **2.6 [2.3 - 3.3]** |
| M2DN (Meng et al., 2024) | 65.2 [49.2 - 76.6] | 3.0 [2.4 - 4.6] | 62.2 [49.2 - 63.3] | 4.6 [4.2 - 5.7] |
| Ours | **73.6 [54.4 - 81.3]** | **2.3 [1.5 - 4.0]** | **77.6 [67.6 - 84.4]** | **2.4 [1.7 - 3.6]** |
| Fully (Behboodi et al., 2022) | - | - | **73.4 [67.5 - 75.7]** | **2.4 [2.2 - 2.4]** |
| Expert | - | - | 84.2 [83.3 - 84.8] | 1.5 [1.0 - 1.6] |

**Training Parameters:** The models are trained for 1000 epochs with a batch size of $B = 16$. Models were trained on an A100 40GB GPU. Full training details are available in (Dorent et al., 2026).

