# OpenReview forum: "Learning with Missing Data: A Multimodal Hierarchical Variational Auto-Encoders for Medical Image Synthesis"
_MIDL.io/2026/Short_Papers — MIDL 2026 - Short Papers Poster_

### Official Review · Reviewer_Qwxk · 2026-04-24
**Multimodal VAE for Cross-Modality Synthesis**

**Rating:** 4
**Confidence:** 4

**Review:**

See strengths and weaknesses.

**Summary:**

This short paper introduces a recently published journal article by the author. It focuses on a variational autoencoder framework for cross-modality image synthesis, with the notable capability of handling missing data during both training and testing phases. The journal paper itself was published in IEEE Transactions on Pattern Analysis and Machine Intelligence on October 2025.

**Strengths:**

Strengths:
The paper presents comprehensive experimental results, including downstream analyses that effectively demonstrate the practical utility of the proposed method. The ability to learn from incomplete datasets is a highly desirable property, particularly in real-world applications where missing data is common.

**Weaknesses:**

Weaknesses:
	1.	The title “Learning from Missing Data” may suggest that the method explicitly leverages the information contained in the missingness itself, whereas the paper primarily focuses on handling incomplete multimodal inputs. Learning from incomplete data would be more appropriate.
	2.	Likely due to space limitations, some important methodological details are insufficiently explained. For instance, the statement that “each product-of-experts component has a factorization similar to the true posterior and can be computed from any subset of observed modalities using only unimodal encoders” would benefit from a more formal presentation, such as with supporting equations.

**Justification Of Rating:**

The paper presents a recently published journal article by the authors. The topic is relevant and likely to be of interest to the MIDL community.

---

### Decision · Program_Chairs · 2026-05-08

Accept (Poster)